# Promising Nature-Based Solutions to Support Climate Adaptation of Arizona's Local Food Entrepreneurs and Optimize One Health

**Yevheniia Varyvoda** [1,*], **Taylor Ann Foerster** [2], **Joona Mikkola** [3] 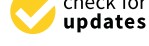 **and Matthew M. Mars** [4]

[1]  Mel and Enid Zuckerman College of Public Health, The University of Arizona, Tucson, AZ 85724, USA
[2]  College of Social & Behavioral Sciences, The University of Arizona, Tucson, AZ 85719, USA; tfoerster@arizona.edu
[3]  Arid Lands Resource Sciences, The University of Arizona, Tucson, AZ 85719, USA; joonamikkola@arizona.edu
[4]  College of Humanities, The University of Arizona, Tucson, AZ 85721, USA; mmars@arizona.edu
**\***  Correspondence: varyvoda@arizona.edu

**Abstract:** This study explores the uptake and potential application of nature-based solutions (NbS) that are particularly promising for small-scale farmers, ranchers, and food entrepreneurs operating in arid and semi-arid regions. Studying the adoption of NbS by local food entrepreneurs (LFEs), including related strengths and limitations, remains an area of exploration due to their potential to optimize interventions that foster environmental sustainability at the intersection of people, animals, and natural ecosystems (i.e., One Health). A multi-method design was used, including literature review, questionnaires, and semi-structured key informant interviews to assess adaptation needs and NbS among a sample of LFEs located in Southern AZ, USA. The findings revealed that existing NbS have been introduced mostly through learning-by-doing practices that are bounded by economic and technological resource constraints. The paper describes a range of accessible approaches and practices that can be piloted and/or scaled up to enhance local food system resilience and contribute to the overlapping health of people, animals, and natural ecosystems. The priority adaptation pathways for NbS were identified to be funding and financing and the co-creation and sharing of knowledge through peer-to-peer and expert-to-peer approaches. The results suggested that AZ LFEs are likely to adopt NbS based on their capacity to address priority climate-driven issues, revenue generation potential, and seamless augmentation with existing food production and operational activities.

**Keywords:** nature-based solutions; local food systems; entrepreneurs; climate change; community innovation



## 1. Introduction

Studies have increasingly demonstrated that climate change has considerable impacts on food systems in arid and semi-arid regions, compromising the intersectional health of humans, domestic and wild animals, plants, and surrounding natural ecosystems, i.e., the One Health dynamic [1]. Arizona (AZ) is the sixth-largest state in the United States (US). AZ is located in the southwestern corner of the country where it is especially challenged by rapid population growth, shifting land use and land cover, increasingly scarce water resources, and consequently, long-term drought [2]. Indeed, much of AZ is characterized as arid to semiarid, with annual average precipitation ranging from less than 4 inches in the desert lowlands to around 40 inches in the mountain highlands. The deserts are some of the hottest and driest areas of the country, while the higher terrain of the Colorado Plateau has much colder winters and relatively mild summers [3]. AZ has warmed by about 2.5 °F (1.35 °C) over the last century. In the past two decades, AZ has experienced its highest recorded temperatures. Phoenix, AZ, had its all-time warmest summer on record in 2023, with an average summer temperature of 97.0 °F (36.11 °C), surpassing the previous record of 96.7 °F (35.94 °C) established just two years earlier in 2020 [4].

AZ is one of the states in the US that has been hardest hit by climate change to the detriment of human health and wellbeing, economic prosperity and social welfare, natural and modified ecosystems, food and water security, and cultural heritage. Symptoms of said climate change include rising air temperatures; changes to the timing, form, and amount of precipitation and associated flooding events; increases in extreme heat events; surface and groundwater reductions; heat stress and heat-related illnesses; and intensifying dust storms and wildfires [5,6]. Likewise, the economic and environmental sustainability and community impacts of AZ's local food systems are continually threatened by climate change. AZ's agri-food industry, valued at 23 billion USD, is currently facing significant climate-driven threats, with 90% of the state having been affected by "exceptional drought" over the past two decades [7].

Small-scale farmers, ranchers, and food purveyors who operate locally across AZ, whom together we refer to hereafter as local food entrepreneurs (LFEs), are confronted with drops in production and sales rates and operational inefficiencies that are perpetuated by socioeconomic constraints and climate uncertainties. As a result, LFEs are forced to downsize production (e.g., removing orchard trees and multi-year crops, selling off portions of herds and flocks), resulting in lost profits and declines in the availability and affordability of locally produced, highly nutritious foods [8].

Beyond the financial consequences, climate change also affects the physical heath and wellbeing of the LFEs. Specifically, climate change, and especially heat stress, can exacerbate a range of acute conditions that include heat stroke, dehydration, and exhaustion, as well as longer term cardiovascular and water- and zoonotic-borne diseases [9]. Further, the financial uncertainties that are being intensified by climate change make LFEs more prone to anxiety, depression, and other debilitating mental health challenges that can, in the extreme, lead to suicide [10].

The overall effects of climate change undermine the health and wellbeing of LFEs and the economic, environmental, and social sustainability of AZ's local food systems. Unfortunately, this wicked dynamic is repeated in countless other locales and regions, making AZ a highly relevant living laboratory in which to further describe the economic, environmental, and social impacts of climate change on local food systems, as well as to explore potential interventions. Climate adaptation interventions tailored to AZ's local and regional needs with health-enhancing benefits are urgently required to meet the climate-driven challenges and optimize the overlapping wellbeing and vibrancy, i.e., One Health, of people, animals, and ecosystems.

Here, we use a multi-method, single case study design to explore the potential effects of NbS on climate change adaptations carried out by LFEs and the One Health of the locales and regions they serve. The International Union of Conservation of Nature (ICUN) Global Standard defines NbS as "actions to protect, sustainable manage and restore natural and modified ecosystems in ways that address societal challenges effectively and adaptively, to provide both human well-being and biodiversity benefits" (IUCN Global Standard for NbS, 2020, p. 1) [11]. NbS for climate adaptation comprise a broad portfolio of interventions with particular attention being paid to fostering natural regeneration and restoration with native species [12]. Given the lack of consensus on which agricultural practices are considered NbS, there are no data available on how widely NbS are used in agriculture at any scale [13]. Accordingly, our current objective is to generate insights into the experiences and everyday realities of LFEs that stand to inform NbS development and implementation. We draw on the insights generated to propose nature-based climate adaptation strategies aimed at enhancing food system resilience.

## 2. Materials and Methods

We created a single case study that was designed to assess the likely viability of NbS in the Southern AZ local food system as viewed through the lenses of LFEs who are actively engaged in food production, storage and distribution, processing, retail, and marketing. For our purposes, Southern AZ was bounded to four neighboring counties:

Cochise, Pima, Pinal, and Santa Cruz. Geographically, these counties are mostly rural and largely agriculture-based, though a large majority of the 1.65 million residents live in Pima County within the Tucson metroplex area [14]. Northern Pinal County stretches into the suburban outskirts of the Phoenix metroplex and is home to nearly 500,000 of the residents within the four-county study region. The rural–urban–suburban mix that characterizes this bounded area is consistent with the understanding that local food systems rely on short supply chains that typically connect rural production areas with urban and suburban consumption sites [15].

The research approach mobilized multiple disciplines (community development, communications, public health, sociology) and spanned a diverse array of food system activities (production, processing, distribution, consumption) and health, socioeconomic, and environmental outcomes. Data were generated through semi-structured interviews and questionnaires that together explored and examined the experiences and everyday realities that shape Southern AZ LFEs' perspectives on climate change, climate adaptation, One Health dynamics, and ultimately the relevancy and viability of NbS.

*2.1. Qualitative Method: Interview Protocol*

2.1.1. Sampling and Data Collection

Interviewees were purposefully selected using maximum variation and chain sampling strategies [16,17]. To foster maximum variation in the sample, we limited our selection to a single general criterion: active as an LFE in the Southern AZ local food system. The LFE role included producers, processors, and purveyors as previously indicated [18]. The sampling strategies generated a sample composed of 11 Southern AZ LFEs.

Each interview began by asking the participating LFE a series of demographic questions, including age, gender, level of education, relevant experience in agriculture or food production and distribution, and current activities as an LFE. From there, the LFE was asked questions pertaining to their current and anticipated climate related challenges, mitigation strategies used to cope with climate challenges, and familiarity with NbS. Thereafter, each interview was concluded with a series of questions aimed at identifying the need for resources, barriers to adoption of NbS, and perceived opportunities surrounding NbS. The interviews were conducted either in-person or over Zoom, audio recorded, and transcribed verbatim for analysis.

2.1.2. Qualitative Data Analysis and Trustworthiness

The interview data were analyzed inductively using a grounded, open-code approach to reveal initial codes and emergent trends [19]. Multiple rounds of ideographic and nomothetic analysis were then conducted, refining initial patterns and trends into salient themes [20]. Two forms of triangulation were used to enhance the trustworthiness of the findings [21,22]. First, researcher triangulation was conducted throughout the analytical process, which involved the members of the research team consistently discussing, debating, and reaching consensus on the insights as they emerged from the data. Second, data triangulation was used to compare, affirm, refine, and further contextual the findings developed from the interviews with those generated from the questionnaire described below. Further, the transferability of the qualitative elements of the findings are enhanced by presenting them in the Results and Discussion sections using rich, thick descriptions [23].

2.1.3. Quantitative Method: Questionnaire Design

The questionnaire was designed to gain insights into the constructs of the Theory of Planned Behavior (TPB). TPB posits that attitude, social norms, and perceived behavioral control predict behavioral intention [24]. Thus, the current research used these constructs to explore baseline data of the potential motivators of LFEs' intentions to adopt NbS into their practices.

We used an online questionnaire to capture participant demographic information and describe existing behaviors and attitudes pertaining to behavior, perceived social norms,

and perceived behavioral control beliefs intentions, and perceptions of beneficial resources regarding the use of NbS in participants LFEs' practices.

Participants completed an online questionnaire consisting of 25 questions that spanned demographic-, psychological-, and resource-related items. Questions were designed in line with the TPB constructs [23] to gauge participant attitudes, perceived social norms, and perceived behavioral control, along with their intention to engage with NbS. Participants then ranked resources from most to least beneficial to the implementation of NbS in their own practice. A non-random convenience sample (*n* = 16) was generated by recruiting participants through researcher networks and online local food resource sites. Participants were required to be an adult working in the production, distribution, aggregation, processing, packaging, and/or retail selling of local food within the Southern AZ local food system (as previously defined). Data collection occurred between October 2023 and February 2024.

The questionnaire consisted of 26 questions and took participants on average around eight minutes to complete. The questionnaire was assessed for face validity through an expert review and all constructs were adapted from previous measures as indicated in the respective construct descriptions. Cronbach's alpha coefficients were used to determine construct reliability (Table 1). Unless otherwise noted, all variables were measured using a five-point Likert-type scale with scores ranging from one (strongly disagree) to five (strongly agree). If participants were not actively engaged with growing food items themselves, they were asked to provide their perspectives in relation to the products that they source.

**Table 1.** Cronbach's alpha reliability estimates by questionnaire constructs.

| Construct | Cronbach's Alpha |
| --- | --- |
| Existing behavior | 0.96 |
| Attitudes | 0.96 |
| Perceived social norms | 0.88 |
| Perceived behavioral control | 0.83 |
| Behavioral intention | 0.96 |

Demographics. At the beginning of the questionnaire, participants were asked about the following demographics: age, gender, highest level of education and discipline of degree if relevant, years of experience in the industry, and the industry that they were currently a part of (i.e., food production, aggregation and distribution, processing and packaging, retail, other).

Defining NbS. After measuring demographic information [25,26], participants were asked to read the following definition of NbS:

"Nature-based solutions are actions to protect, sustainably manage, or restore natural or modified ecosystems to address societal challenges, simultaneously providing benefits for people and the environment [27]. Solutions include protection or conservation of natural areas, soil health practices, water management, agroforestry, afforestation using native tree species, or other practices that contribute to climate change adaptation while improving the health and livelihoods of farmers, ranchers, and food entrepreneurs".

Following the definition, participants were asked to indicate that they had read the definition to proceed with the questionnaire.

Existing behavior. To measure participant's current engagement with NbS practices, two items were used. These items were rated using a five-point Likert-type scale to indicate frequency, with a score of one indicating that a participant never used NbS and a score of five indicating that participants always used NbS practices. The items included "In my current practice, I use NbS" and "I incorporate nature based solutions into my current activities as a local food entrepreneur".

Attitude. Participants' attitudes towards NbS were rated using three different five-point Likert-type scales adapted from Miller et al., (2019) [25]. Specifically, attitudes towards NbS were portrayed as a function of effectiveness of NbS, evaluations of NbS, and favorableness of using NbS. Each scale was proportionate to the other, with a score of one

indicating NbS were "not at all" the quality of focus (i.e., effective, good, favorable) and a score of five indicating "Extremely" the quality of focus. The same item was used for each scale "For me, using NbS would be...".

Perceived social norms. Perceptions of social norms as they relate to NbS were measured using four items that were adapted from Miller et al. (2019) and Senger et al. (2017) [25,26]. The items captured both injunctive (two items) and descriptive (two items) norms to create the construct of social norms. Example items included "Most people who are important to me approve of the use of NbS" and "Most people who are important to me incorporate NbS in their practice".

Perceived behavioral control. The extent to which participants perceived their control over the adoption of NbS was measured using four items adapted from Miller et al., (2019) [25]. Example items included "If I wanted to, I could implement NbS" and "Using NbS in my practice is in my control".

Behavioral intention. Participant's intention to adopt NbS for themselves was measured using three items adapted from Senger et al., (2017) [26]. Example items included "I plan to use NbS in the future" and "I intend to adopt NbS in the next three years".

Resources. Participants were asked to rank four resources: (1) Funding, (2) Community (i.e., connections with other local food producers or related community resources), (3) Education, and (4) Additional labor (i.e., volunteers, employees) along with an "other" option with the opportunity to explain within the textbox provided under the option. Specifically, participants ranked the resources in order of most beneficial to least beneficial to them if they were to implement NbS in their own LFE activities. The resources ranked were informed by data from the interview portion of this research.

## 3. Results

In this section, we present the results of the questionnaire. We then use the Discussion section to contextualize and bring nuance to the said results through the integration of the insights generated from the analysis of the qualitative interview data.

### 3.1. Questionnaire Participant Demographics

The questionnaire sample ($n$ = 16) consisted of more women (68.8%) than men (31.3%). The participant age range was from 25 to 69 years of age with an approximate mean age of 41 years old. The distribution of the sample by highest level of education obtained was as follows: high school education (6.3%), associate degrees (12.5%), bachelor's degree (37.5%), master's degrees (12.5%), a doctorate (6.3%), and other (e.g., some wrote college; 25%). Participants indicated that within their respective industries they had between one year and 26 years of experience. On average, participants had approximately seven years of experience. When asked about their current position as an LFE, 68.8% of participants selected food production, 31.3% food aggregation and distribution, 56.3% food processing and packaging, and 56.3% selected retail.

### 3.2. Descriptive Results

Given the low sample size ($n$ = 16), descriptive statistics were used to provide baseline insights into the perceptions surrounding the adoption of NbS among the surveyed LFEs. All statistical analyses were performed using SPSS software (IBM Corp. Released 2023. IBM SPSS Statistics for Macintosh, Version 29.0.2.0 Armonk, NY, USA: IBM Corp). The results revealed that participants on average used (or sourced products from suppliers that use) NbS about half of the time in their current practice. Attitudes ($M$ = 4.02) about using NbS were positive on average, with a score of four indicating very good, favorable, or effective evaluations of using NbS. Perceptions of social norms ($M$ = 3.59) and behavioral control ($M$ = 3.52) averaged slightly positive scores, with a score of three for both constructs indicating a neutral position (Neither agree not disagree) and a score of four indicating agreeing somewhat with the statements presented. Behavioral intention to adopt NbS was rated relatively high ($M$ = 4.29), indicating that, on average, participants held positive

intentions to engage with NbS in their own practice in some capacity. Table 2 provides the scores for each construct as well as the standard deviation.

**Table 2.** Descriptive statistics by construct.

| Construct | M | SD |
|---|---|---|
| Existing behavior | 3.16 | 1.31 |
| Attitudes | 4.02 | 1.16 |
| Perceived social norms | 3.59 | 0.79 |
| Perceived behavioral control | 3.52 | 0.89 |
| Behavioral intention | 4.29 | 0.95 |

Participants rated constructs on five-point Likert-type scales. The existing behavior items used a five-point frequency scale ranging from "never—one" to "always—five". The items used to measure attitude were rated on a five-point scale to provide an evaluative perception of NbS with a score of one indicating "not at all" the quality of focus (i.e., effective, good, favorable) and a score of five indicating that using NbS were rated as "Extremely" effective, good, or favorable. All other constructs were measured using a scale of agreement, such that higher scores indicate more agreement with the statements and thus higher levels of the construct.

*3.3. Resources*

Thirteen of the sixteen participants ranked the resources provided (i.e., funding, community, education, labor, and an other option) from most beneficial to least beneficial to them for implementing NbS in their LFE activities. Table 3 summarizes the rankings of the resources. Overall, Funding was ranked as the most beneficial resource more frequently than the other resources. Funding was followed by Community (i.e., connections with other local food producers or related community resources), which was most frequently rated as the second most beneficial resource by participants. One participant provided "equity share" as another resource. Resources were ranked from most beneficial (first) to least beneficial (fourth). The "other" option provided consistently ranked as fifth.

**Table 3.** Results of ranking resources for implementing NbS.

| Resource | First | Second | Third | Fourth |
|---|---|---|---|---|
| Funding | 9 | 4 | | |
| Community | 3 | 6 | 3 | 1 |
| Education | 1 | 2 | 6 | 4 |
| Additional labor | | 1 | 4 | 8 |

## 4. Discussion

### 4.1. Current Uptake of NbS

Over 40 NbS have been identified as evidence-based practices that are known to be effective in mitigating and adapting to high-priority climate-related issues identified by LFEs [22,28–61] (Supplementary Materials, Table S1). This list is not exhaustive and NbS such as cover crops and diversified crop rotations are already widely used practices in sustainable farming. US farmers are undertaking five broad categories of climate adaptation strategies based on NbS: water management, crop management, nutrient management, technological management, and financial management [62]. Our study has shown that many small-scale farmers, ranchers, and other LFE-types are familiar with and practicing NbS. However, they might not be fully aware of the benefits of One Health and limitations associated with NbS. For example, one participant stated, "I've heard the term, but I know a lot of people have different descriptions of it. That sounds like my standard farming practices". As such, the information presented can help LFEs to leverage specific NbS that

are likely to deliver desirable climate adaptation outcomes. Table 4 summarizes coping strategies used by AZ local food entrepreneurs to address climate-driven threats.

**Table 4.** Coping strategies used by AZ local food entrepreneurs to address climate-driven threats.

| Theme | Coping Strategies in Place to Address Climate-Driven Threats |
|---|---|
| Water management | Irrigation (Olla irrigation, drip irrigation, micro irrigation, irrigation scheduling, targeted irrigation), water harvesting and reuse system, dry farming, planting heritage crops, deep watering. |
| Crop management | Changing crop varieties, diversification in crops, wildlife habitat improvement, microclimate creation, planting crops in raising beds or pots, crop switching, crop rotation, creating the windbreaks, pollinators habitat creation, shifted planting dates. |
| Nutrient management | Cover crops, reduced or zero tillage, customized compost, organic fertilizers, rotational grazing, livestock integration on the field, biochar application, mulching, use of microorganisms for dry soil restoration. |
| Low-Tech management | Swamp cooling, evaporative coolers, optimization of energy efficiency, optimization of logistics, using the mechanical equipment that nature-positive (a roller crimper for no till planting), shading nets, moisturizing of the shading nets, hail nets, on-farm data collection, organza fruit netting bags. |
| Financial management | Developing a regenerative cuisine, collaboration with federal agencies and cooperative extension professionals. |

There are numerous examples of NbS that are being successfully funded, permitted, and implemented across regions and sectors by many of the US' federal agencies and partners [17]. However, the full-scale successful adoption of NbS is influenced by financial and operational barriers that impede any stage of the implementation cycle of NbS. Overcoming said barriers is reliant on the level of awareness and perceptions of climate change and existing adaptation strategies by LFEs and the financial accessibility and longer term performance of the involved solutions.

In AZ, LFEs are critical stakeholders in and facilitators of NbS adoption. After analyzing the perceptions, preferences, and perspectives of Southern AZ LFEs on NbS, we identified that a lack of access to funding and financing, limitations in available labor and expertise, and insufficient awareness as principal barriers to NbS adoption. Such insights lead us to reinforce three main pathways are NbS adoption. Table 5 summarizes the main factors for NbS adoption among Southern AZ LFEs as reflected in the relevant literature and across our interview data.

**Table 5.** Enablers of and barriers to NbS adoption among AZ local food entrepreneurs.

| Enablers | Barriers | References |
|---|---|---|
| Pathway 1: Funding and Financing of NbS | | |
| Availability of national and local funding and financing drawn from public, private, and alternative sources to support the implementation and management of NbS.<br>Strong consumer interest in locally and sustainably produced foods represents an important opportunity for greater financial support of LFEs in NbS adoption.<br>The potential of NbS to create jobs and increase agriculture output and food security, while also contribute to climate adaptation and environmental and human health benefits. | The perceived high expenses associated with NbS implementation and maintenance. Farmers are unlikely to adopt climate-smart agricultural practices or actions unless they bring greater profitability compared to their current practices or if there are incentives provided. Limited awareness of available payment mechanisms for privately implemented NbS.<br>The divide between landownership and farming activities can pose challenges for LFEs when it comes to long-term planning and investments on the farm. Additionally, the pressure for private land development and increasing land values can restrict access to forage and create obstacles for generational transfer and the adoption of new management practices.<br>Farmers and ranchers are implementing various strategies such as reducing the size of their herds, purchasing hay, and investing in water conservation technologies to address the challenges of declining revenues and increasing debts.<br>Existing market structures are not favorable towards nature-positive food production.<br>There is currently no study on farmer's interest in alternative sources to finance climate adaptation practices i.e., crowdfunding. | [5,11,13,22,62,63] |
| Pathway 2: Peer-to-peer (P2P) and Expert-to-peer (E2P) knowledge co-creation and exchange | | |
| LFEs possess a keen awareness of weather patterns and climatic conditions that affect them on a daily basis. They continuously adapt to changes in both short-term and long-term weather and climate variability.<br>The AZ Cooperative Extension organization plays a critical role in bringing research and innovative approaches to local-level stakeholders in all fifteen counties and from five Native nations.<br>The exponential rise in informational resources and consulting services available about NbS.<br>Understanding the value and benefits of NbS for climate adaptation and resilience.<br>It is very likely that increased use of NbS practices, e.g., in relation to heat mitigation or weed and pest control, will reduce food entrepreneurs' exposure to chemicals and thus contribute to reducing the large number of associated negative acute and long-term health impacts.<br>Over the years AZ's farmers and ranchers have successfully cultivated crops and raised herds of livestock in areas that are prone to drought and have increasing precipitation variability. It has equipped food producers in the region with a wealth of inter-generational knowledge, enabling them to adapt to changing climate conditions.<br>As signs of climate change become apparent, ranchers and farmers are more engaged in having conversations about the issue, including its risks and possible solutions.<br>Local food entrepreneurs' interest in NbS adoption. | Finding the specific information and tailored NbS food entrepreneurs need can be challenging due to the large amount of content available.<br>Lack of a scientific understanding of NbS and their benefits and limitations.<br>Limited knowledge/information on NbS tailored for small-scale food production in the Southwest.<br>Within the food storage and distribution elements of the food systems, NbS are not considered as relevant or applicable.<br>The length of time and labor intensity needed for NbS practices to deliver concrete benefits is uncertain.<br>The riskiness of changing practices. Food producers prefer to implement reactive adaptation strategies that are informed by their previous experiences.<br>Aging farming population.<br>The effectiveness of implemented NbS is not well quantified.<br>The population in Arizona continues to increase and residential areas continue expand, which leads to an additional demand for and reliance on natural resources. | [5,62,64–66] |

*4.2. NbS Adaptation Pathways*

Pathway 1: Funding and Financing of NbS. NbS are recognized as cost-effective responses to climate change and environmental degradation that can simultaneously benefit One Health outcomes [67]. There are numerous examples of NbS that can support improved One Health outcomes (e.g., desert agroforestry, biochar application, rotational grazing, photo-selective shading made of natural materials). Such outcomes include reducing diseases, improving food security and nutrition, enhancing mental health, mitigating heat stress, improving soil health, and more. By protecting, conserving, restoring, and sustainably managing natural or modified ecosystems, NbS serve as essential tools in climate change adaptation, promoting biodiversity, ensuring clean air and water access, and strengthening food security [68].

Even when aware of the multiple benefits provided by NbS, LFEs remain reluctant to adopt said solutions due to having a limited understanding of the financial implications, specifically when it comes to who should bear the expenses and potential funding sources [69]. Further, small-scale farmers prioritize immediate financial returns over climate-smart agricultural practices because of the high risks and low returns associated with agricultural production [36]. Therefore, they require compensation for the environmental benefits they provide, assistance with acquiring affordable and less labor-intensive solutions, and access to the various financial resources needed to adopt climate-smart and regenerative farming techniques [70]. According to research conducted by Hovis et al. (2023) in eastern North Carolina, the primary factors influencing the willingness to accept payments for NbS are landowners who are younger, possess more wealth, and manage larger parcels of land [71].

NbS are typically small, stakeholder-driven initiatives tailored to local environmental and socio-economic conditions, as well as culture, values, and traditions [72]. Across the European Union, 72% of projects based on NbS cover less than 1 km$^2$ or 100 ha [73]. Returning to AZ, NbS are trending towards newer, smaller, and more urban operations; from 1997 to 2017, the average farm size in Arizona decreased by 57% while the number of farms increased by 127% [74]. Nearly one quarter (24%) of farmers are considered beginner farmers, meaning they have 10 years of operational experience or fewer [74,75].

Further consideration of funding and financing for NbS adoption in AZ agri-food sector is especially important given nearly 50% of farms have less than ten acres of land in their operation, with 88% of these same farms generating less than 25,000 USD in annual sales [76]. Indeed, AZ farmers and LFEs associate high costs with the implementation and maintenance of NbS as evidenced by the following participant quotes:

- "I don't think the problem is that farms don't have the right amount of information to solve problems. The problem is they don't have the resources to do it. And resources just in capital".
- "We also need to be able to get more money to the farmer, the farmer earns about seven cents out of every food dollar it spends and that's not enough for the farmer to be able to adapt to climate change of any type".

This perception is largely perpetuated by the lack of quantitative evidence for the benefits, costs, and overall effectiveness associated with NbS implementation and maintenance for climate adaptation. As an example, the integration of NbS with desert agroforestry and agricultural land and habitat restoration would require 1 to 1.5-acre feet (AF) per acre of water, as compared to an average of 4.5-AF-per-acre of water for conventionally grown AZ crops. Habitat restoration would use even less water if and when native desert vegetation is established [49]. Importantly, not all NbS have the same financial implications and are thus dependent on local and regional variations [77].

AZ's small-scale farmers and LFEs face a significant challenge in identifying and accessing funding and financing sources for the implementation of NbS. They lack a comprehensive understanding of the payment options available to them to incentivize NbS. Payment is defined as the funding, financing, and/or partnership mechanisms utilized in the implementation and management of NbS [78]. More specifically, funding mechanisms

include grants and donations provided by federal, state, and philanthropic sources to support a one-time cost for specific NbS projects. Unlike loans, these funds do not need to be repaid by the recipient. While some programs may have match requirements for the recipients, they often waive or reduce those requirements for economically disadvantaged communities. This enables them to access the necessary funds without facing excessive financial burdens. Financing mechanisms such as loans and bonds can provide the supplementary project funds that are needed. However, they require repayment and interest for their use [79].

There are a variety of public, private, and alternative sources of funding and financing available to AZ's small-scale farmers and LFEs, though user awareness and the time required to seek and access said sources are remarkably low.

NBS projects generate a multitude of public benefits such as food and water security, disaster risk reduction, climate change mitigation and adaptation, reversing ecosystem degradation and biodiversity loss, human health, and socio-economic development. However, high sunk costs coupled with the lack of immediate revenue can pose a challenge to implementing NbS [80]. Furthermore, NbS often rely on public financing due to previously identified barriers like low revenue potential and susceptibility to market failures [67].

Public investments play a significant role in funding and financing NbS projects, they account for up to 82% (165 billion USD) of the funding allocated to NbS-related projects on a global scale. Over 71% (117 billion USD) of public finance for NbS is allocated to biodiversity and landscape protection and to sustainable agriculture, forestry, and fishing [81].

Federal Funding: To date, there are 140 federal funding programs offered by a diverse set of agencies that are currently supporting or have the potential to support NbS. These programs have primarily focused on providing extensive assistance for the planning, design, implementation, and construction stages of NbS, while operations, maintenance and monitoring are less supported [17]. Table 6 lists examples of funding opportunities offered by federal programs that could provide assistance to AZ's small-scale farmers, ranchers, and food entrepreneurs who are interested in the adoption of NbS.

There is a detailed guide to federal programs for sustainable agriculture, forestry, entrepreneurship, conservation, food systems, and community development that local small-scale farming and food entrepreneurs can use to leverage the existing resources for NbS implementation [82].

Financial support for NbS initiatives can be provided under various state-funded programs. For example, Arizona Department of Agriculture through the Resilient Food Systems Infrastructure program funding will help to build resilience in Arizona's local and regional food systems. The proposed program would assist small-scale farms and ranches, new and beginning farmers and ranchers, underserved producers, veteran producers, and underserved communities in the realization of infrastructure projects to make food production, aggregation, and processing more efficient. It will contribute to establishing more stabilized markets for producers and expand capacity for supplying culturally appropriate food through emergency food systems and programs [83]. These programs include the Graham & Greenlee County Micro-Loan Fund and the Arizona Natural Resources Conservation Service (NRCS), among others.

Private financing: According to R.C. Brears (2022, p. 34), private finance for NBS is the raising, provision, or management of private capital to conserve, restore, sustainably use, or avoid a negative footprint on biodiversity and ecosystem services. Private finance includes commercial banks and investment companies, developers and infrastructure operators, and private equity and infrastructure funds [84]. Approximately 18% (equivalent to 35 billion USD) of the total financial resources allocated to NbS come from private financing worldwide. In the United States, farmers contribute around 390–450 million USD through private investment to conservation agriculture [81].

**Table 6.** Examples of funding opportunities offered by federal programs [17] that could provide assistance to AZ's small-scale farmers, ranchers and food entrepreneurs in the adoption of NbS.

| Agency | Funding Program | The Types of NbS That Are Eligible for Support | NbS project Lifecycles Supported by the Program * | | | |
|---|---|---|---|---|---|---|
| | | | 1 | 2 | 3 | 4 |
| USDA | Conservation Reserve Program (CRP) (https://www.fsa.usda.gov/programs-and-services/conservation-programs/conservation-reserve-program/index Accessed on 14 June 2023) | "Environmentally sensitive agricultural land conservation. Long-term, resource-conserving plant species such as approved grasses or trees (known as "covers") to control soil erosion, improve water quality, and develop wildlife habitat". | X | X | X | X |
| USDA | Emergency Conservation Program (https://www.fsa.usda.gov/programs-and-services/conservation-programs/emergency-conservation/index Accessed on 14 June 2023) | "Rehabilitation of farmland and conservation structures damaged by natural disasters and implementation of emergency water conservation measures in periods of severe drought". | X | X | – | – |
| USDA | The State Acres for Wildlife Enhancement (SAFE) Initiative (https://www.fsa.usda.gov/Assets/USDA-FSA-Public/usdafiles/FactSheets/archived-fact-sheets/state_acres_wildlife_enhancement_init_jul2015.pdf Accessed on 15 June 2023) | "Soil conservation, water quality protection, or wildlife habitat enhancement". | X | X | X | X |
| USDA | The Agricultural Conservation Easement Program (ACEP) (https://www.nrcs.usda.gov/programs-initiatives/acep-agricultural-conservation-easement-program Accessed on 15 June 2023) | "Protection of croplands and grasslands on working farms and ranches by limiting non-agricultural uses of the land through conservation easements". | X | X | – | X |
| USDA | Conservation Innovation Grant (CIG) (https://www.nrcs.usda.gov/programs-initiatives/cig-conservation-innovation-grants Accessed on 15 June 2023) | "The development of new tools, approaches, practices, and technologies to further natural resource conservation on private lands". | X | X | X | X |
| USDA | Conservation Stewardship Program (CSP) (https://www.nrcs.usda.gov/programs-initiatives/csp-conservation-stewardship-program Accessed on 15 June 2023) | "Development and implementation of practices and activities that expands on the benefits of cleaner water and air, healthier soil and better wildlife habitat, all while improving agricultural operations". | X | X | X | – |
| USDA | Environmental Quality Incentives Program (https://www.nrcs.usda.gov/programs-initiatives/eqip-environmental-quality-incentives Accessed on 26 June 2023) | "Technical and financial assistance to improve water and air quality; conserve ground and surface water, increase soil health; reduce soil erosion and sedimentation; improve or create wildlife habitat; mitigation against drought and increasing weather volatility". | X | X | X | X |
| USDA | Landscape Conservation Initiatives (https://www.nrcs.usda.gov/programs-initiatives/landscape-conservation-initiatives Accessed on 26 June 2023) | "Voluntary on-farm conservation initiatives to improve water and air quality, soil health, and preserve the wildlife habitats". | X | X | X | X |

**Table 6.** *Cont.*

| Agency | Funding Program | The Types of NbS That Are Eligible for Support | NbS project Lifecycles Supported by the Program * | | | |
|--------|----------------|------------------------------------------------|---|---|---|---|
| | | | 1 | 2 | 3 | 4 |
| USDA | National Water Quality Initiative (NWQI) (https://www.nrcs.usda.gov/programs-initiatives/national-water-quality-initiative Accessed on 26 June 2023) | "Voluntary on-farm conservation initiatives that promote soil health, reduce erosion and lessen nutrient runoff, such as filter strips, cover crops, reduced tillage and manure management". | X | X | X | X |
| USDA | Regional Conservation Partnership Program (RCPP) (https://www.nrcs.usda.gov/programs-initiatives/rcpp-regional-conservation-partnership-program Accessed on 26 June 2023) | "To support the adoption of climate-smart agriculture practices, which have direct climate mitigation benefits, advance a host of other environmental co-benefits, and offer farmers, ranchers and foresters new revenue streams". | X | X | – | – |
| USDA | Environmental Quality Incentives Program—WaterSMART Initiative (https://www.nrcs.usda.gov/programs-initiatives/eqip-watersmart/priority-areas Accessed on 26 June 2023) | "To increase water conservation and resilience to drought. Water conservation improvements may also improve soil health; reduce soil erosion, sediment, nutrient, and pathogen losses; protect crop health and productivity; and make using equipment, facilities, and agricultural operations more efficient". | X | X | X | X |

* NbS project phases are 1—planning and design; 2—implementation or construction; 3—operations and maintenance; 4—monitoring (X—supported; – not supported).

The private-sector production, storage, distribution, and consumption components of the agri-food system have financial interests in achieving a more sustainable food production and consumption in Arizona [5]. These can include: (a) mid-size and large-scale farmers, ranchers, beef, dairy, and poultry producers operating with sustainability approaches; for example, Duncan Family Farms, Oatman Farms and Oatman Flats Ranch, Ramona Farms, the San Xavier Cooperative Farm; (b) agribusiness, especially innovators such as AZ Baking Company, Barrio Bread, the Food Conspiracy among others; (c) agricultural and conservation associations, such as Arizona Farm Bureau, Arizona Beef Council, Dairy associations, Southwest Black Ranchers, Agribusiness & Water Council of Arizona, Western Growers, Arizona Grain Research and Promotion Council, The Arizona Association of Conservation Districts, Ajo Center for Sustainable Agriculture, Baja Arizona Sustainable Agriculture, The Yuma Center of Excellence for Desert Agriculture, etc. Potential partners and investors can be engaged among the various actors that are engaged in food system activity.

However, it can be challenging for small-scale farmers and LFEs to navigate and secure federal and state public funding [71]. So, encouraging them to engage in NbS projects requires access to and the availability of alternative financing models. According to den Heijer C and Coppens, "Alternative financing is arrangements that draw on financial resources other than public budgets collected through general taxation. They could be sourced from institutional investors, businesses, and citizens" [67].

AZ's food producers and LFEs may consider various alternative financial models that they can apply when it comes to adopting NbS. Carbon markets. Carbon markets, albeit still immature, could become a significant source of funding for NbS. Different carbon market mechanisms vary from compliance programs, such as the European Union's emission-trading scheme (EU-ETS) and California's Cap-and-Trade program, to voluntary and international carbon market places such as the Voluntary Carbon Market (VCM) and Internationally Transferred Mitigation Outcomes (ITMOs) under the Paris agreement article 6.2.

There are four major carbon registries operating in the United States: ACR (formerly known as American Carbon Registry), Climate Action Reserve (CAR), Verra's Verified Carbon Standard (VCS), and Gold Standard. These four registries have issued more than 412 million credits corresponding to 412 million metrics tons of carbon dioxide ($CO_2$) equivalent to projects based in the United States for both voluntary and compliance markets [85,86].

Although some carbon market programs do not yet allow carbon offsets to be purchased from farmers, their role in funding sustainable food system transformation and NbS is likely to grow. The global carbon credit market demand was estimated at 181 million metric tons of $CO_2$ equivalents in 2023 and is expected to grow to 1.2 billion metric tons by 2030 and 5.4 gigatons by 2050 [87]. According to a study by McKinsey, the demand for carbon credits could grow by a factor of more than 15 by 2030 and by a factor of 100 by 2050 [88].

Commercialization of NbS initiatives: There are various biomass sources available, including crops grown specifically as biomass inputs, the leftovers from crop and wood harvesting, and manures, all of which can contribute to small-scale biochar production. Biochar technology can improve soil productivity and capture carbon. The integration of biochar technology with small-scale farm operations is especially promising given the associated esource constraints and available labor. Moreover, biochar technologies can increase water and fertilizer efficiencies, enhance the re-purposing of organic wastes, and improve crop productivity [89]. With the growing availability of lower cost commercial biochar equipment, such innovation stands to make small-scale farmers more environmentally and economically sustainable.

Philanthropy and charity funding: Partnerships with philanthropic organizations and private philanthropists who have an interest in NbS for adapting to climate change should be considered as a potential source of additional funding to supplement the resources

provided by government agencies, private entities, and others [90]. There is a growing number of these foundations that focus on community-based sustainable agriculture, aiming to combat climate change, improve access to healthy and nutritious food, promote regenerative agriculture, and build local food systems resilience. Examples of such foundations include the Rockefeller Foundation, the Foundation for Food and Agriculture, the Gordon & Betty Moore Foundation, the Walton Family Foundation, the Howard G. Buffett Foundation, The Schmidt Family Foundation 11th Hour Project, IKEA Foundation, The David & Lucille Packard Foundation, the Bill and Melinda Gates Foundation, the Mars Wrigley Foundation, The Cedar Tree Foundation, etc.

Crowdfunding: Crowdfunding became widely popular thanks to the rise of social media and online platforms dedicated to directly connecting entrepreneurs with backers (i.e., crowd-funders, investors), such as crowdfunding websites [91]. There have been no studies conducted on the interest of LFEs in Arizona in using alternative finance options, such as crowdfunding, to support NbS. However, a study by Kragt et al. (2021) revealed that farmers with a financially secure farming business, previous exposure to crowdfunding, or a strong sense of responsibility to mitigate climate change are more interested in using crowdfunding [91].

Household/business investment: According to Burton and Otte (2022), the importance of being innovative and having a willingness to invest has been recognized as key drivers for successfully implementing climate mitigation and adaptation measures at the family farm level [92]. There is a progressive trend of declining climate mitigation/adaptation intentions throughout the life-cycle stages of the farm as a business (investment, debt, expansion, consolidation, contraction, exit) and due to generational aspects (early-stage developers, commercial developers, commercially disengaged and semi-retired withdrawers). For example, in the case of semi-retired withdrawers, promoting new investment in the farm may pose challenges due to the population of aging farmers and their low intentions to invest. The average age of food producers in Arizona is 59.4 years. With this insight in mind, we propose that, to effectively promote, design, and implement NbS while leveraging various funding sources, it is important to consider the life cycle of agri-food entrepreneurial activity.

Rebates, tax credits, or low-interest financing can incentivize NbS adoption. For instance, the Rainwater Harvesting rebate program, sponsored by Tucson Water, offers rebates of up to 2000 USD to individuals or small businesses who install a rainwater-harvesting system. Qualifying options include passive rainwater harvesting, which involves directing and retaining water in the landscape, as well as active rainwater harvesting, such as the use of storage tanks for future water usage [93].

Although there are various alternative financial models available for NbS adoption, none of them can completely substitute traditional public finance [33]. Yet, NbS' adoption in small-scale food systems have the potential to enhance individual farm's productivity, efficiency, and resilience while also delivering broader societal benefits like climate change mitigation, biodiversity, water conservation, and One Health support. It is essential to identify all potential private and public benefits that are linked to the implementation of NbS project in order to unlock funding sources.

Pathway 2: Peer-to-peer (P2P) and Expert-to-peer (E2P) knowledge co-creation and exchange: providing interactions in both the real world and online settings. Understanding the sources of information about NbS that LFEs and experts/advisors rely on and trust is valuable for scaling the adoption of NbS. This knowledge can help identify the most effective providers to disseminate reliable and trustworthy information about NbS. AZ small-scale farmers and LFEs have emphasized the value of peer-to-peer and expert-to-peer knowledge exchange. Examples of quotes from interview participants include the following:

- "It's being done around the world. It's being done right next door and you just don't know it".
- "So to have somebody to walk with the farmer, and say we're going to do this together".

- "To look more towards interpersonal relationships, whether that be a consultant, a neighbor, but that these relationships need to be more consistent, more streamlined and more accessible to everyone".

Peers are found to offer valuable guidance and assistance in the practical adoption of NbS due to their previous experiences with similar challenges e.g., loss of food production, reduced soil quality, variation in temperature and precipitation, and increased pests. This prevents costly trial-and-error learning and encourages the exploration of underutilized or neglected solutions for common problems.

Rust et al. (2022) found that, in a sample of farmers from Hungary and the UK, that participants placed the most trust in other farmers (i.e., peer-to-peer) when learning about new soil practices. In contrast, participants revealed that they were less trusting of institutions and research outside of the fields of agriculture [64].

Experts are found to offer valuable guidance and assistance in developing field-ready solutions, introducing innovative affordable climate smart approaches for water conservation, novel fertilizers, pathogen control, and real-time field monitoring.

According to Sutherland et al., (2017) small-scale farmers utilize three main types of networks to access different types of knowledge. One type is a centralized network, where farmers interact individually with an advisor who then interacts with other advisors. Distributed networks represent communities or networks of practice and involve peers exchanging personal knowledge to varying degrees. Decentralized networks involve knowledge from sources outside the peer group and are associated with acquiring potential knowledge, such as future or cutting-edge innovations. The structure of these networks reflects the farmers understanding of available sources of knowledge, and their credibility [94].

To ensure successful adoption and scaling-up of NbS within small-scale food systems, it is essential to engage all types of knowledge networks. A centralized network can be used to establish one-to-one interactions with formal advisors who can provide valuable information on funding and financing options, can assist in completing grant and loan applications, as well as help develop a compelling business case for NbS in small-scale agri-food systems. Additionally, distributed networks should be utilized to facilitate collaboration and the exchange of knowledge and best practices related to NbS implementation on-site. Lastly, decentralized networks play a key role in promoting advances in knowledge and practices regarding NbS.

Different levels and quality of informational support are needed for the planning, implementation, and stewardship of NbS. By leveraging these various types of knowledge networks, the adoption of NbS can be accelerated among AZ small-scale food producers and entrepreneurs.

The interventions implemented to ensure public health and safety during the COVID-19 pandemic (and now endemic) led to notable increases in the use of digital knowledge-exchange platforms. To stay up to date with this trend it is recommended to develop a peer-to-peer and expert-to-peer knowledge sharing, capacity building, and networking platform, which allows farmers and LFEs to connect and interact with each other, as well as with local subject matter experts to facilitate and accelerate the adoption of NbS, increasing Arizona's small-scale food systems resilience.

Although there is a growing availability of resources and information on the incorporation of NbS, there remains a lack of a user-friendly guide for practitioners, specifically LFEs [79]. This guide should contain a set of procedures to accompany each phase of NbS projects, including planning and design, implementation or construction, operations and maintenance, and monitoring. Having access to such a document would greatly support LFEs in developing and implementing evidence-based NbS and help to avoid maladaptation during the process.

To bridge the implementation gap, the eLearning training course "Advancing Arizona's agrifood systems climate adaptation through NbS was developed. It aims to improve awareness and understanding of the science, policy, practice, resourcing, and governance

of NbS specific to Arizona, and support an adoption of NbS by small-scale food system actors to boost food and nutrition security and to strengthen the resilience of food systems. The course incentivizes continual learning about locally available and accessible NbS that work best to (i) enhance food security and the nutrition of individuals, households, and communities; (ii) benefit human, animal, and environmental health; (iii) promote climate resiliency; and (iv) generate economic viability. The course showcases a range of available, proven, accessible NbS that can be piloted and/or scaled up to enhance food security and the health and wellbeing of LFEs across Arizona.

*4.3. Promoting the Adoption of NbS*

The adoption of NbS relies on the synergy between stakeholders (e.g., local farmers, ranchers, LFEs), experts from many disciplines (e.g., public health, agriculture, food systems, environment), and affected communities, creating a value-added and participatory approach that can give rise to context-driven implementable solutions. Although the scientific community and policymakers recognize and promote the importance of NbS in climate adaptation, their practical implementation is struggling mainly due to a lack of awareness among the main stakeholders.

Currently, NbS have primarily been introduced through learning-by-doing practices and LFES actively seeking implementable solutions while working within economic and technological resource constraints. Limited finances, time, and labor resources, needs for infrastructural improvements, the availability and effectiveness of extension services and outreach, uncertainty regarding implementation processes and their effectiveness are the most frequently cited unmet needs of AZ's local food system stakeholders in response to climate change with NbS. Various factors, such as encouragement for NbS initiatives, recognition of diverse expertise, and the significance of incorporating different knowledge and practice-based perspectives, can impact the level to which LFEs will adopt NbS. AZ's LFEs are likely to adopt NbS based on their capacity to address priority climate-driven issues, revenue generation potential, and seamless augmentation with existing food production and operational activities. These key factors will play a critical role in the adoption process.

Based on a literature review, consultations with subject-matter experts and targeted outreach to stakeholders, the following gaps need to be bridged to promote the adoption of NbS among LFEs in Arizona.

- The need to analyze available private and alternative sources of funding and financing. The resource guide will increase the connectivity among potential investors, LFEs, and policymakers across the local food system and enable the expansion of investments to support and accelerate the adoption of NbS. Evidence-based and context-based quantification of the benefits, costs, and cost-effectiveness of NbS implementation and maintenance for climate adaptation are required. This information will help to mobilize financial resources and generate investible business cases to accelerate the implementation of NbS.
- The need to raise awareness among AZ's LFEs. It is important to emphasize that NbS can be integrated and effectively applied at various stages within the food system, not limited to just agricultural production. NbS hold the capacity to benefit various elements of the food system, including food production, storage and distribution, processing and packaging, retail and marketing. Farmers, processors, distributors, transporters, and retailers whose livelihoods, health, and wellbeing are threatened by climate change can gain advantages from NbS.
- The need to document on-farm/on-ranch coping strategies used by AZ LFEs and quantify the performance of NbS in addressing climate-driven threats. This includes assessing the benefits and challenges associated with NbS implementation.
- The need for support in the planning, implementation, and stewardship of NbS. In order to achieve substantive and lasting results, small-scale farmers, ranchers, and LFEs need to be supported and navigated by Cooperative Extension professionals.

Close collaboration between LFEs and experts is needed to accelerate the adaptation of NbS and prevent wasted efforts on impractical or infeasible solutions. To achieve this, it is necessary to develop professional learning resources aimed at enhancing AZ's Cooperative Extension's capacity to transfer knowledge regarding NbS adoption at the local/state level.

Taking into consideration the existing uncertainties, this study provides AZ's local small-scale farmers, ranchers, and LFEs with the information and tools they need to immediately and proactively implement NbS with One Health benefits (e.g., use of microorganisms for dry soil restoration, construct nesting boxes to conserve raptors and reduce virus risk, apply multi-story cropping for desert adapted crops, etc.).

Our findings and recommendations are aligned with those observed in other regions across the world. They emphasize the critical role of finding and financing, knowledge and information, revenue generation, and the capacity for the seamless integration of NbS with existing food production and operational activities. This is essential for small-scale farmers, ranchers, and LFEs who are considering NbS implementation.

*4.4. Study Limitations*

A non-random sampling technique was used to conduct this research, limiting the generalizability of the results which remain limited to those who participated. Given that research on perceptions of NbS is currently scant, the current research provides valuable contributions to the understanding of this topic despite its limitations.

The sample size was relatively small, and this limits our ability to conduct advanced statistical relationship testing. Future research should inquire into perceptions about NbS with a larger sample size to identify key relationships between variables. For example, relationships between attitudes, social norms, perceived behavioral control, and intention, as described in the Theory of Planned Behavior [24]. Additionally, the current research looked at NbS as a whole, and disaggregating the concept into specific NbS would be helpful in understanding the adoption of NbS more comprehensively.

**5. Conclusions**

The study explores the implementation potential of NbS that are particularly promising for supporting climate adaptation among LFEs and carry the capacity to optimize One Health in Arizona. It reveals current climate-related challenges experienced by Southern AZ's LFEs who promote and support the local food systems through localized production, distribution, and consumption. The study also explores the coping mechanisms that have already been implemented, adaptation needs, and NbS that can be better harnessed to address food security and deteriorating health outcomes within arid and semi-arid regions such as AZ using available and accessible NbS. A total of over 40 promising NbS were identified and proposed that could advance Arizona's small-scale food system's resilience and optimize One Health in a context-specific and cost-effective manner. Small-scale farmers, ranchers, and LFEs have various options for adopting NbS, depending on their specific needs and available resources. To prevent maladaptation, it is necessary to generate additional evidence regarding NbS' value and limitations to address climate adaptation and sustainable balance and optimize the health of people, animals, and ecosystems. The information and tools herein hold the potential for replication and scaling by both practitioners and researchers across other arid and semi-arid regions, particularly in low- and middle-income communities and countries because they address food security and deteriorating health outcomes with high-value, cost-effective interventions arising from available and accessible ecosystem services.

**Supplementary Materials:** The following supporting information can be downloaded at: https://www.mdpi.com/article/10.3390/su16083176/s1, Table S1: Promising nature-based and climate-smart solutions to support climate adaptation of Arizona's LFEs and optimize One Health.

**Author Contributions:** Conceptualization, M.M.M., Y.V. and T.A.F.; methodology, M.M.M. and T.A.F.; formal analysis, T.A.F. and Y.V.; investigation, T.A.F. and Y.V.; resources, M.M.M. and Y.V.; data curation, T.A.F.; writing—original draft preparation, Y.V., T.A.F. and J.M.; writing—review and editing, M.M.M.; supervision, M.M.M.; project administration, Y.V.; funding acquisition, Y.V. and M.M.M. All authors have read and agreed to the published version of the manuscript.

**Funding:** One Health Research Initiative, The University of Arizona Health Sciences: 2023-One Health Initiative Pilot Grant.

**Institutional Review Board Statement:** The project was approved in June 2023 by the Institutional Review Board, The University of Arizona, USA.

**Informed Consent Statement:** All participants gave their informed consent for inclusion before they participated in this study.

**Data Availability Statement:** The original contributions presented in the study are included in the article/Supplementary Materials, further inquiries can be directed to the corresponding author.

**Conflicts of Interest:** The authors declare that they have no known competing financial interests or personal relationships that could have appeared to influence the work reported in this paper.

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
