# Peer review of "Promising Nature-Based Solutions to Support Climate Adaptation of Arizona’s Local Food Entrepreneurs and Optimize One Health"

_sustainability, doi:10.3390/su16083176_

Round 1

Reviewer 1 Report

Comments and Suggestions for Authors

The study is very up-to-date and based on a broad methodological basis. The consequences of climate change are not only ecological, but also have an economic and social effect on society. The research explores opportunities to deploy nature-based solutions that are particularly promising for supporting climate adaptation among local small-scale farmers, ranchers and food entrepreneurs. Over 40 promising NbS have been identified that can improve the sustainability of Arizona's small-scale food system and optimize One Health in a specific and cost-effective way.  Overall, the introduction is too one-sided, and includes only 6 sources. It is necessary to expand and the consequences of climate change to be indicated (generally). This also applies to the "Results" section - the obtained data should be described and analyzed in more detail. The Material and Methods and Discussion sections are very well presented and developed. As a result of the conducted research, valuable information is obtained, which provides an opportunity to understand and identify important problems, as well as to make adequate and correct decisions.

Author Response

Dear Reviewer,

Thank you for the time and thought you gave to our initial submission. We greatly value the time and effort you put intoreviewing the manuscript and have taken your feedback to heart. In doing so, we believe that doing so has greatly improved our intended contribution. Below we outline our responses to each specific point raised by each reviewer. We were highly attentive to reducing any potential, unintentional plagiarism flags throughout the revision. Though, we welcome any further insights into any lingering issues and will gladly address any of remaining concerns. As a writing team, we are adamantly committed to the highest of ethical standards. We appreciate your support in this regard.

Sincerely,

The Writing Team

Reviewer 2 Report

Comments and Suggestions for Authors

Abstract: Hardly any real content in it, and needs a lot of revision or even a complete rewrite. For example, mixed methods were what? What's the sample size, or what it is n=? “will” should be changed to the past tense. For international journal, many readers do not know where is AZ, and why you choose the place as arid and semi-arid region? Use abbreviations the first time you mention nature-based solutions, and abbreviations throughout. Abstract should be a high summary of the entire manuscript. Maybe, add more background at the start, such as relationship on One Heath and NbS.

Introduction: add more introduction on other’ methods and results relative with similar goals of your research.

M&M: why AZ why you choose the place as arid and semi-arid region? Give more information on rainfall, temperature in tested year, longitude, latitude, etc.

Fig 1: This drawing is too rough. There should be no background lines.

CO2 equivalent, 2 shoud be in state of lower case.

For all abbreviations, use abbreviations the first time you mentioned, and abbreviations throughout.

Conclusions: Please summarize it highly, condensing it into 2 paragraphs or even 1 paragraph.

Comments on the Quality of English Language

72 of 100

Author Response

(The authors gave the same response as above.)

Reviewer 3 Report

Comments and Suggestions for Authors

Dear Authors,

I had the opportunity to read and review the manuscript entitled „Promising nature-based solutions to support climate adaptation of Arizona’s local food entrepreneurs and optimize One Health”.

This study aims to generate insights into the experiences and everyday realities of local food entrepreneurs that stand to inform nature-based solutions (NbS) development and implementation.

In my opinion, the research topic is absolutely in the scope of the journal's focus, the study identifies practical NbS for climate adaptation but also promotes a collaborative and participatory approach to implementation, helping to tailor solutions to local contexts and contribute to sustainability goals. My review below suggests some improvements.

Title: informative and consistent with the study's content.

Abstract: the abstract provides a clear and reasonable overview of the study. A summary of the research's objectives, methods, and findings is provided.

Keywords: the title already contains all of them, so they need to be reconsidered.

1. Introduction: by emphasizing the impact of climate change on local food systems, the connection between climate adaptation and health, NbS as a solution, and the research objectives, the introduction effectively highlights the originality and novelty of the article.

2. Materials and Methods: in my opinion, the sample of 16 respondents for the quantitative research is not sufficient to obtain reliable results. The definition of the target population - adults who work in the production, distribution, aggregation, processing, packaging, or retail of local food in Southern and Central Arizona regions - does not justify the small sample size. This study uses only the constructs of Theory of Planned Behavior (TPB), but does not examine their relationships.

3. Results – Questionnaire: the section is not numbered, only results from the survey are provided.

4. Discussion: the discussion section provides the results of the qualitative data analysis. A sample consisting of 11 Southern AZ local food entrepreneurs was used for qualitative analysis. This section also provides examples of funding opportunities offered by federal programs that could assist AZ small-scale farmers, ranchers, and food entrepreneurs who are interested in the adoption of NbS.

5. Conclusions: the section numbering is wrong. In its current form, the Conclusion section fails to fulfill its intended purpose: the conclusion does not equal the summary or recommendations. Comparisons with findings from previous studies would have been interesting.

The manuscript is long and lacks conciseness. Providing results should be structured appropriately.

Author Response

(The authors gave the same response as above.)

Round 2

Reviewer 2 Report

Comments and Suggestions for Authors

The quality of the revised version was improved a lot. The results section is a little short in the Abstract. You can add some data or others. Agreed to publish.

Author Response

Dear Reviewer,

Thank you for taking the time to give us your valuable feedback and recommendations. We greatly appreciate your input and gladly address the provided suggestion.

Respectfully,

The Writing Team